# Near-atomic structures of the BBSome reveal the basis for BBSome activation and binding to GPCR cargoes

Shuang Yang[1†], Kriti Bahl[2†], Hui-Ting Chou[1‡], Jonathan Woodsmith[3], Ulrich Stelzl[3], Thomas Walz[1]*, Maxence V Nachury[2]*

[1]Laboratory of Molecular Electron Microscopy, The Rockefeller University, New York, United States; [2]Department of Ophthalmology, University of California San Francisco, San Francisco, United States; [3]Department of Pharmaceutical Chemistry, Institute of Pharmaceutical Sciences, University of Graz and BioTechMed-Graz, Graz, Austria

**Abstract** Dynamic trafficking of G protein-coupled receptors (GPCRs) out of cilia is mediated by the BBSome. In concert with its membrane recruitment factor, the small GTPase ARL6/BBS3, the BBSome ferries GPCRs across the transition zone, a diffusion barrier at the base of cilia. Here, we present the near-atomic structures of the BBSome by itself and in complex with ARL6$^{GTP}$, and we describe the changes in BBSome conformation induced by ARL6$^{GTP}$ binding. Modeling the interactions of the BBSome with membranes and the GPCR Smoothened (SMO) reveals that SMO, and likely also other GPCR cargoes, must release their amphipathic helix 8 from the membrane to be recognized by the BBSome.

*For correspondence:
twalz@rockefeller.edu (TW);
maxence.nachury@ucsf.edu (MVN)

[†]These authors contributed equally to this work

Present address: [‡]Department of Therapeutic Discovery, Amgen Inc, South San Francisco, United States

Competing interests: The authors declare that no competing interests exist.

## Introduction

Cilia dynamically concentrate signaling receptors to sense and transduce signals as varied as light, odorant molecules, Hedgehog morphogens and ligands of G protein-coupled receptors (GPCRs) (*Anvarian et al., 2019*; *Bangs and Anderson, 2017*; *Nachury and Mick, 2019*). Highlighting the functional importance of dynamic ciliary trafficking, the appropriate transduction of Hedgehog signal relies on the disappearance of the GPCR GPR161 and the Hedgehog receptor Patched 1 from cilia and the accumulation of the GPCR Smoothened (SMO) within cilia (*Bangs and Anderson, 2017*). Regulated exit from cilia represents a general mechanism to redistribute signaling molecules on demand (*Nachury and Mick, 2019*). Patched 1, GPR161, SMO and other ciliary membrane proteins are all ferried out of cilia in a regulated manner by an evolutionarily conserved complex of eight Bardet-Biedl Syndrome (BBS) proteins, the BBSome (*Nachury, 2018*; *Wingfield et al., 2018*). While GPR161 and other ciliary GPCRs such as the Somatostatin receptor 3 (SSTR3) are removed from cilia by the BBSome only when they become activated, SMO undergoes constitutive BBSome-dependent exit from cilia in unstimulated cells to keep its ciliary levels low. Accumulation of SMO in cilia is then, at least in part, achieved by suppression of its exit (*Milenkovic et al., 2015*; *Nachury and Mick, 2019*; *Ye et al., 2018*).

Membrane proteins travel into, out of, and within cilia without utilizing vesicular intermediates and remain within the plane of the ciliary membrane (*Breslow et al., 2013*; *Chadha et al., 2019*; *Milenkovic et al., 2009*; *Ye et al., 2018*). Thus, membrane proteins that enter and exit cilia must cross the transition zone (TZ), a diffusion barrier at the base of cilia, by lateral transport (*Garcia-Gonzalo and Reiter, 2017*). Recently, we found that regulated TZ crossing of GPR161 is enabled by the BBSome in concert with the ARF-like GTPase ARL6/BBS3 (*Ye et al., 2018*), but the mechanism

of facilitated TZ crossing by the BBSome remains a fundamental unanswered question (*Nachury and Mick, 2019*).

Our recent cryo-electron microscopy (cryo-EM) structure of the BBSome revealed that the BBSome exists mostly in an auto-inhibited, closed conformation in solution and undergoes a conformational change as it is recruited to membranes by ARL6$^{GTP}$ (*Chou et al., 2019*). Given that ARL6$^{GTP}$ triggers polymerization of a membrane-apposed BBSome/ARL6 coat (*Jin et al., 2010*) and enables BBSome-mediated TZ crossing (*Ye et al., 2018*), the ARL6$^{GTP}$-bound BBSome conformation represents the active form of the complex. Here we determine high-resolution structures of the BBSome alone and bound to ARL6$^{GTP}$, and we map the BBSome–SMO interaction to model how the membrane-associated BBSome–ARL6$^{GTP}$ complex recognizes its cargoes. Surprisingly, our studies reveal that SMO must eject its amphipathic helix 8 (SMO$^{H8}$) from the inner leaflet of the membrane in order to be recognized by the BBSome. Sequence analysis suggests that this may be a general principle for the interaction of the BBSome with its cargo GPCRs.

## Results

### High-resolution structure and model of the BBSome

Following on our previous strategy (*Chou et al., 2019*), we purified the BBSome to near-homogeneity from retinal extract and analyzed its structure by single-particle cryo-EM. The advent of higher throughput direct detector cameras and faster automated data-collection procedures combined with improvements in data processing with new tools implemented in RELION-3 (*Zivanov et al., 2018*) led to a BBSome map at an overall resolution of 3.4 Å from an initial dataset of 770,345 particles (*Figure 1—figure supplement 1*, *Figure 1—figure supplement 2A*).

The BBSome is composed of 29 distinct domains characteristic of sorting complexes (*Figure 1A*). α-solenoids, β-propellers, pleckstrin homology (PH) and appendage domains are all present in multiple copies and our previous map made it possible to build a Cα backbone model that encompassed 25 out of 29 domains (PDB-Dev accession PDBDEV_00000018; *Chou et al., 2019*). For building the current model, the previous Cα model was docked into the map, and the higher-resolution map enabled us to confidently assign side chains for most regions (*Figure 1B*). The new map allowed us to build the coiled-coil domains of BBS1 and BBS9, for which densities were not well-defined in the previous map. Altogether, 27 out of 29 domains distributed across the 8 BBSome subunits could be modeled. Despite the increased resolution of the current density map, the gamma-adaptin ear (GAE) domains of BBS2 and BBS7 could not be modeled, and side chains could not be assigned for BBS2$^{βprop}$, BBS2$^{cc}$, BBS7$^{βprop}$ and BBS7$^{cc}$.

### High-resolution structure of the BBSome bound to ARL6$^{GTP}$

Consistent with our previous observations based on a 4.9 Å resolution map of the BBSome (*Chou et al., 2019*), the new BBSome structure cannot accommodate binding to ARL6$^{GTP}$. Fitting a homology model of the bovine BBS1$^{βprop}$–ARL6$^{GTP}$ complex (based on the X-ray structure of the *Chlamydomonas* complex; *Mourão et al., 2014*) in either BBSome structure caused a steric clash between ARL6$^{GTP}$ and a region encompassing BBS2$^{βprop}$ and BBS7$^{cc}$. These data support a model in which the BBSome exists in an autoinhibited form in solution and undergoes a conformational opening upon recruitment to membranes by ARL6$^{GTP}$, similar to other sorting complexes such as COPI, AP-1 and AP-2 (*Cherfils, 2014*; *Faini et al., 2013*).

The membrane-associated form of the ARL6$^{GTP}$-bound BBSome represents its active conformation, because ARL6$^{GTP}$ enables TZ crossing (*Ye et al., 2018*). To determine the nature and consequence of the conformational change in the BBSome that takes place upon ARL6$^{GTP}$ binding, we set out to determine the structure of the BBSome–ARL6$^{GTP}$ complex.

Mixing recombinant ARL6$^{GTP}$ together with the purified BBSome allowed for complex formation in solution. The BBSome–ARL6$^{GTP}$ complex was analyzed by cryo-EM (*Figure 2—figure supplement 1*), yielding a density map at an overall resolution of 4.0 Å (*Figure 1—figure supplement 2A*). Focused refinement of the top and lower lobes of the complex resulted in improved maps of 3.8 Å and 4.2 Å resolution, which facilitated model building (*Figure 2—figure supplement 1*). Even though the apparent overall resolution was nominally not as good as that of the BBSome alone, several domains were better resolved in the density map of the BBSome–ARL6$^{GTP}$ complex (*Figure 1—*

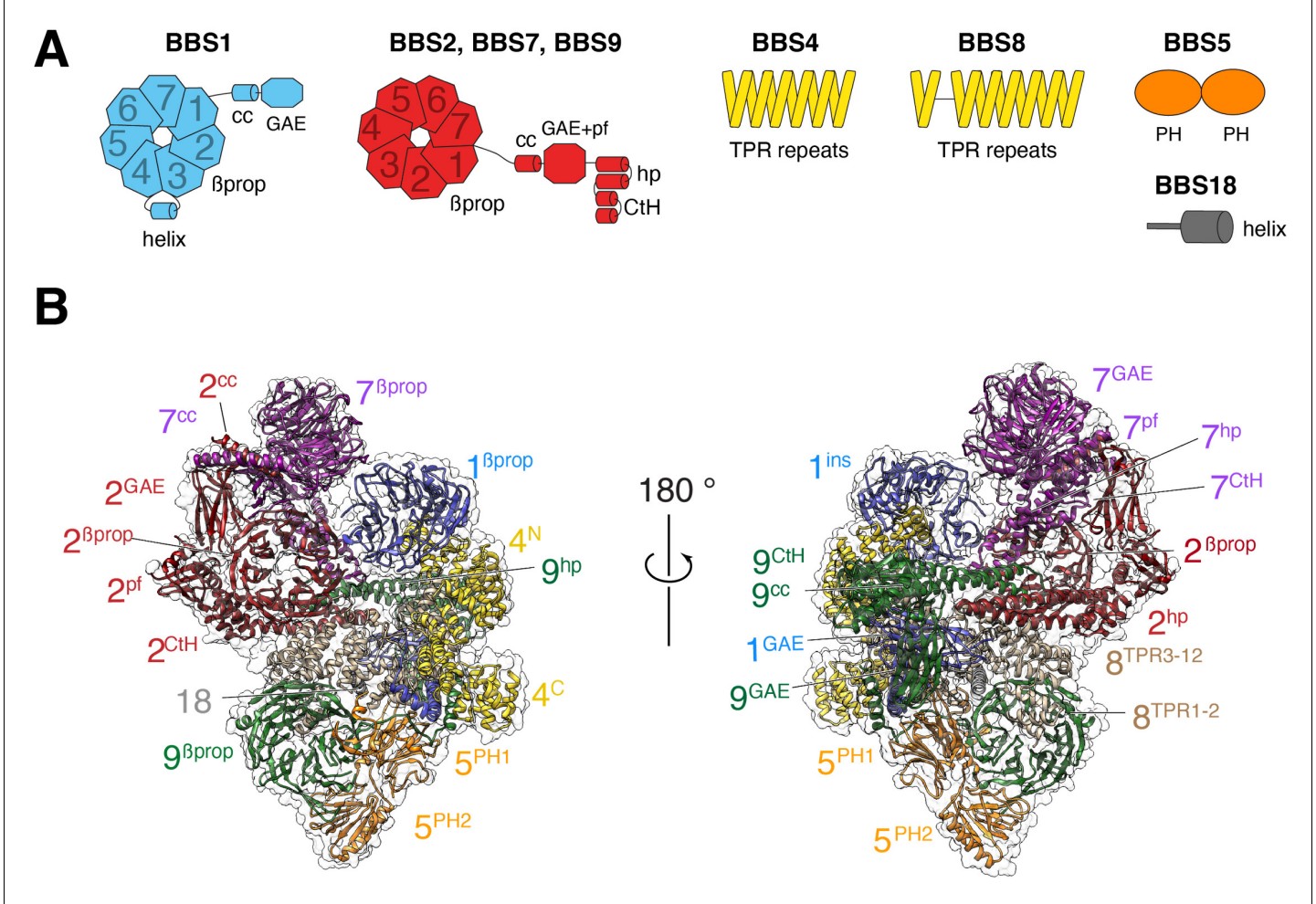

**Figure 1.** Overall structure of the BBSome. (**A**) Diagrams showing the domain architecture of the eight BBSome subunits. βprop, β-propeller; cc, coiled coil; GAE, γ-adaptin ear; pf, platform; ins, insert; hp, hairpin; CtH, C-terminal helix bundle; TPR, tetratricopeptide repeat; PH, pleckstrin homology. (**B**) Two views of the cryo-EM map (transparent surface) and the near-atomic model of the BBSome complex shown in ribbon representation. Individual domains are labeled with the numbers identifying the subunit and the superscripts denoting the specific domain.

The online version of this article includes the following figure supplement(s) for figure 1:

**Figure supplement 1.** Cryo-EM analysis of the BBSome alone.

**Figure supplement 2.** Quality assessment of the BBSome density maps.

*figure supplement 2B*). In particular, the quality of the map was significantly increased for the top β-propeller (*Figure 1—figure supplement 2C*). The improved map quality allowed us to correctly place the β-propellers (βprop) of BBS2 and BBS7, which had been swapped in our previous structural description (*Chou et al., 2019*) due to their extreme similarity and the limited resolution of the previous map. This new assignment is further supported by a recently published structure of the BBSome (*Singh et al., 2020*).

In the BBSome–ARL6 structure, ARL6$^{GTP}$ is nestled in a wedge opening between BBS1$^{βprop}$ and BBS7$^{βprop}$. A ~ 20° rotation of BBS1$^{βprop}$ from the BBSome alone conformation allows ARL6$^{GTP}$ to move away from the steric clash with BBS2$^{βprop}$. This movement of BBS1$^{βprop}$ is accompanied by a twisting of the first two TPR repeats from the BBS4 α-solenoid (*Video 1*), in line with the close association between the N terminus of BBS4 and BBS1$^{βprop}$ seen in the BBSome alone structure and confirmed by cross-link mass spectrometry (*Chou et al., 2019*). Besides the movements of BBS1$^{βprop}$ and BBS4$^{TPR1-2}$, ARL6$^{GTP}$ binding caused only subtle changes in the structure of the BBSome. The movements of BBS4 and BBS1 are in agreement with two recently published structures of the ARL6$^{GTP}$-bound BBSome (*Klink et al., 2020*; *Singh et al., 2020*).

We note that the conformational opening of the BBSome is likely spontaneous as a minor 3D class corresponding to the open form could be detected in our previous dataset of the BBSome alone. As the 3D class of the open conformation contained only a small percentage of the particles in the dataset, the equilibrium between closed and open form in solution is strongly shifted towards the closed form. Binding to ARL6$^{GTP}$ would thus act as a thermodynamic sink that locks the BBSome into the open conformation.

Small GTPases of the ARF/ARL family undergo conformational changes in three regions upon nucleotide exchange from GDP to GTP: the Switch 1 and 2 regions and the Interswitch toggle (*Sztul et al., 2019*). As previously found in the crystal structure of the BBS1$^{βprop}$–ARL6$^{GTP}$ complex (*Mourão et al., 2014*), the BBS1$^{βprop}$ makes contacts with the Switch 2 region and with helix α3 of ARL6$^{GTP}$ while the Switch 1 region of ARL6 is readily available for interacting with other, yet unidentified, complexes (*Figure 2B*). Interestingly, the 'backside' of ARL6$^{GTP}$ (i.e., the surface on the opposite side of the Switch regions) interacts with a loop that connects BBS7$^{βprop}$ and BBS7$^{cc}$. Given the absence of conformational changes in the backside of ARL6 upon nucleotide exchange, ARL6 binding to the BBS7$^{βprop}$-BBS7$^{cc}$ loop will not be gated by the nucleotide state, similar to the proposed binding of ARF1 to the γ subunit of the clathrin adaptor AP-1 (*Ren et al., 2013*).

In addition, the physical interaction between ARL6$^{GTP}$ and the upper lobe removes the upper lobe flexibility previously observed in the BBSome alone preparation, resulting in a more stable BBSome conformation (*Figure 1—figure supplement 2C*).

## A conceptual model for BBSome binding to cargoes and membranes based on mapping of the SMO–BBSome interaction and the cryo-EM structure

To gain insights into how the BBSome ferries its cargoes out of cilia, we sought to model the binding of the BBSome to membranes and cargoes. We started by mapping the interaction of the BBSome with its well-characterized cargo SMO. The BBSome directly recognizes the cytoplasmic tail of SMO that emerges after the seven-transmembrane helix bundle (SMO$^{Ctail}$, aa 543–793; *Klink et al., 2017*; *Seo et al., 2011*) and is required for the constitutive removal of SMO from cilia (*Eguether et al., 2014*; *Goetz et al., 2017*; *Zhang et al., 2011*), p. 3). Using in vitro-translated (IVT) BBSome subunits, we found that BBS7 was the only subunit unambiguously captured by SMO$^{Ctail}$ (*Figure 3A*). BBS7 was also the sole subunit to recognize SSTR3$^{i3}$ (*Figure 3—figure supplement 1A*). Truncations of SMO$^{Ctail}$ revealed that the first 19 amino acids of SMO$^{Ctail}$ are necessary and sufficient for binding to BBS7 (*Figure 3B*). The specificity of BBS7 binding to SMO was retained when BBSome subunits were assayed against the first 19 amino acids of SMO$^{Ctail}$ (*Figure 3—figure supplement 1B*). Systematic yeast two-hybrid (YTH) testing using a collection of well-validated constructs (*Woodsmith et al., 2017*) identified a direct interaction between SMO$^{Ctail}$ and a BBS7 fragment C-terminal to the β-propeller (BBS7[326-672]; BBS7$^{C}$) (*Figure 3C*). Again, deletion of the first 10 amino acids from SMO$^{Ctail}$ abolished the YTH interaction with BBS7$^{C}$ (*Figure 3D*). In close agreement with our findings, BBS7 is one of only two BBSome subunits associating with SMO$^{Ctail}$ in co-IP studies and deletion of the first 10 amino acids of SMO$^{Ctail}$ abolishes the interaction with BBS7 (*Seo et al., 2011*). The congruence of co-IP, YTH and GST/IVT-capture assays strongly supports the conclusion that the first 10 amino acids from SMO$^{Ctail}$ and BBS7 are the major determinants of the SMO–BBSome interaction.

The location of the BBSome-binding determinant on SMO is surprising because the crystal structures of SMO have revealed that the first 10 amino acids of SMO$^{Ctail}$ form a membrane-parallel amphipathic helix termed helix 8 (H8) (*Byrne et al., 2016*; *Deshpande et al., 2019*; *Huang et al., 2018*; *Qi et al., 2019*; *Wang et al., 2014*; *Wang et al., 2013*; *Weierstall et al., 2014*; *Zhang et al., 2017*). We thought to determine how the BBSome recognizes SMO$^{H8}$ by mapping the residues of SMO$^{H8}$ required for association with BBS7. A prior study found that the BBSome recognizes motifs consisting of an arginine preceded by an aromatic residue (*Klink et al., 2017*). One motif in SMO$^{H8}$ is a perfect match (549WR550) and another is a looser candidate (553WCR555). Mutation of both Trp549 and Trp553 from SMO$^{H8}$ completely abolished binding to BBS7, with Trp549 being the major determinant (*Figure 4A*). Similarly, mutation of Arg550 greatly diminished binding to BBS7. We conclude that each residue within the 549WR550 motif contributes to BBSome binding. The direct binding of BBS7 to Trp549 of SMO was unexpected, because all crystal structures of SMO find these residues embedded within the hydrophobic core of the lipid bilayer (*Figure 4B*,

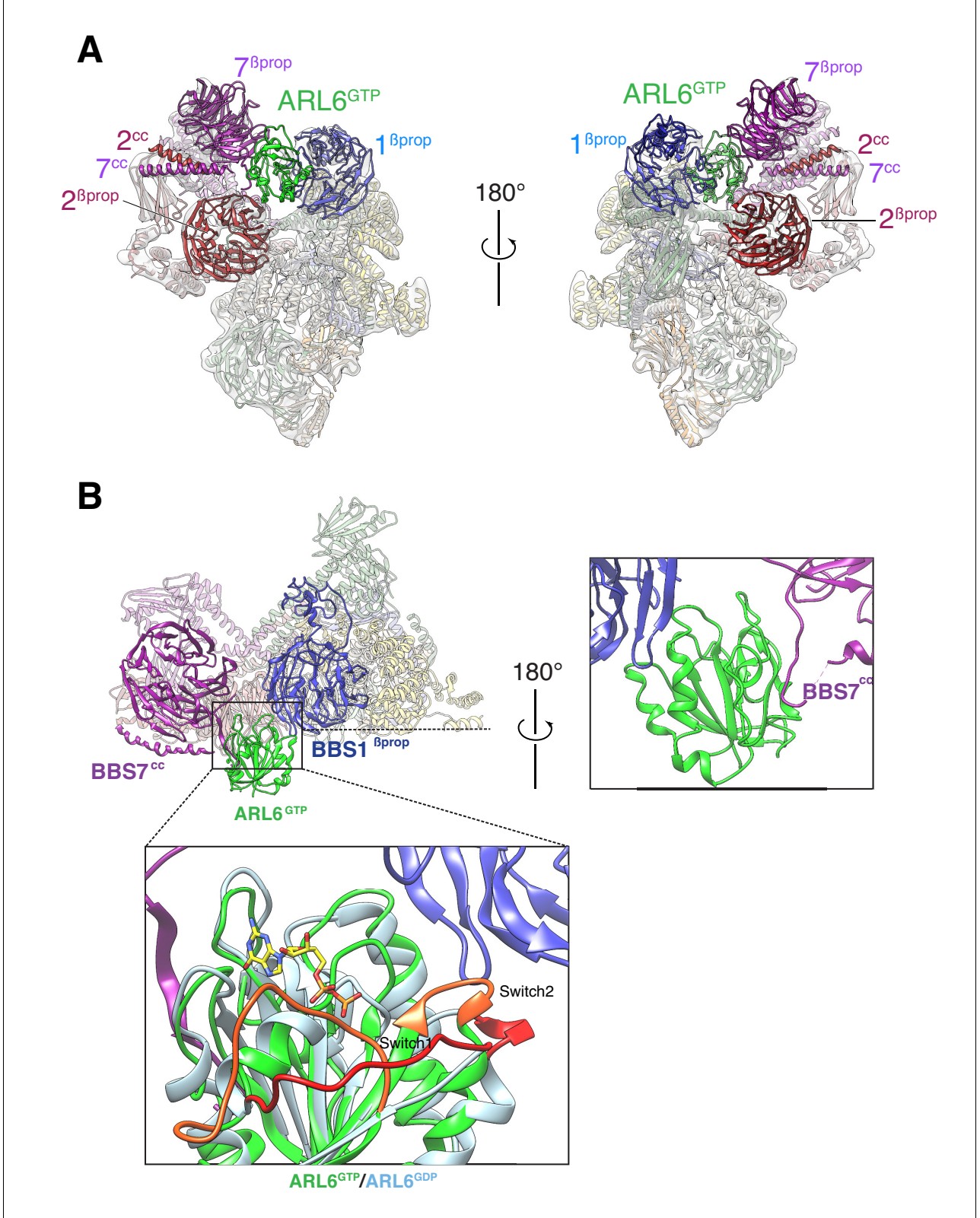

**Figure 2.** Overall structure of the BBSome–ARL6$^{GTP}$ complex. (**A**) Two views of the cryo-EM map (transparent surface) and the near-atomic model of the BBSome–ARL6$^{GTP}$ complex shown in ribbon representation. BBS2$^{\beta prop}$ and BBS7$^{\beta prop}$ were swapped in our previously published structure, as were BBS2$^{cc}$ and BBS7$^{cc}$. BBS1$^{\beta prop}$, the domain that contacts ARL6$^{GTP}$, is also labeled. (**B**) Overall view (left panel) of the BBSome–ARL6$^{GTP}$ complex. Right panel: close-up view focusing on the interaction of the backside of ARL6$^{GTP}$ with the loop connecting BBS7$^{cc}$ and BBS7$^{\beta prop}$, and with BBS1$^{\beta prop}$.

*Figure 2 continued on next page*

Figure 2 continued

Bottom panel: The Switch 1 and Switch 2 regions change conformation between ARL6$^{GDP}$ for ARL6$^{GTP}$ and these regions are colored orange (ARL6$^{GTP}$) or red (ARL6$^{GDP}$). ARL6$^{GTP}$ contacts the BBS1 β-propeller with its Switch 2 region. In contrast, ARL6 contacts BBS7 using a surface that is largely unaffected by nucleotide binding. A homology model of bovine GDP-bound ARL6$^{GTP}$ (based on the crystal structure of the *Chlamydomonas* protein; PDB ID: 4V0K) was aligned to the model of the GTP-bound ARL6 in our BBSome–ARL6$^{GTP}$ complex (ARL6$^{GDP}$ in light blue and ARL6$^{GTP}$ in lime green). GTP is shown in stick representation.

The online version of this article includes the following figure supplement(s) for figure 2:

**Figure supplement 1.** Cryo-EM analysis of the BBSome–ARL6$^{GTP}$ complex.

*Figure 3—figure supplement 1D*). It follows that SMO$^{H8}$ must be extracted from the membrane for the BBSome to recognize SMO.

Amphipathic helices generally fold upon insertion into the membrane and remain as random coil in solution (*Drin and Antonny, 2010*; *Seelig, 2004*). Helix 8 is a near-universal feature of GPCRs (*Piscitelli et al., 2015*) and a peptide corresponding to helix 8 of rhodopsin adopts a helical conformation when bound to membranes but is a random coil in solution (*Krishna et al., 2002*). Such membrane requirements for folding of helix 8 are likely generalizable to other GPCRs (*Sato et al., 2016*). We therefore propose that SMO$^{H8}$ exists in an equilibrium between a folded membrane-embedded and an unfolded state, and that it is the out-of-the-membrane, unfolded state that binds to the BBSome.

ARL6-GTP binding recruits the BBSome to the membrane, and our structure of the BBSome–ARL6 complex (*Figure 2*) together with the binding assays (*Figure 3*) leads to a conceptual model for how the BBSome associates with the membrane to recruit SMO. Further mapping of the SMO$^{H8}$–BBS7 interaction by GST/IVT-capture assays indicated that BBS7$^{βprop}$ was necessary and sufficient for the interaction with SMO$^{H8}$ (*Figure 5A*). In YTH, deletion of BBS7$^{cc}$ specifically abolished the SMO$^{Ctail}$–BBS7$^{C}$ interaction (*Figure 5B*). Thus, while YTH and GST/IVT-capture assays identified different regions of BBS7 recognizing SMO$^{Ctail}$, the regions identified by YTH (BBS7$^{cc}$) and GST/IVT capture (BBS7$^{βprop}$) are adjacent to one another in the BBSome structure. To unify these findings, we propose that an extended SMO$^{H8}$ is recognized by a surface encompassing BBS7$^{cc}$ and BBS7$^{βprop}$.

We conceptualized BBSome binding to membrane and SMO based on our binding studies. Considering that Trp549 of SMO is contacted by BBS7$^{βprop}$ and that the first amino acid of SMO after the 7$^{th}$ transmembrane helix is Lys543, BBS7$^{βprop}$ must be within 6 amino acids or ~21 Å of the membrane. Similarly, BBS7$^{cc}$ must be located within 10 amino acids of the membrane. We note that if SMO$^{H8}$ were to remain helical once extracted from the membrane, BBS7$^{βprop}$ would need to be within 9 Å of the membrane. Because no BBSome–ARL6 orientation can be achieved that brings BBS7$^{βprop}$ within less than 15 Å of the membrane, we conclude that SMO$^{H8}$ must be unfolded to be recognized by the BBSome. Secondly, ARL6$^{GTP}$ anchors the BBSome to the membrane. Because ARF family GTPases bind lipid bilayers through their amphipathic N-terminal helix inserted in a membrane-parallel orientation within the lipid-headgroup layer, the starting point of the core GTPase domain of ARL6 at Ser15 informs the anchoring of the BBSome–ARL6$^{GTP}$ complex on membranes. In the resulting conceptual model of the membrane-associated BBSome–ARL6$^{GTP}$ complex bound to SMO (*Figure 5C*), the orientation with respect to the membrane of ARL6$^{GTP}$ in

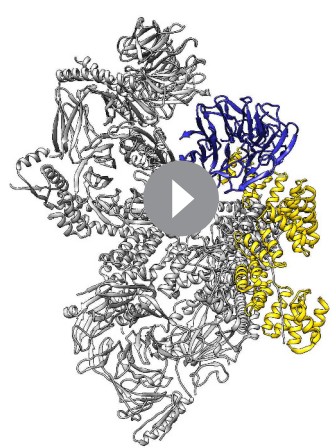

**Video 1.** Morph of the BBSome structure from the unbound to the ARL6$^{GTP}$-bound conformation and back to unbound conformation. BBS1$^{βprop}$ is blue, BBS4 is yellow and ARL6$^{GTP}$ is magenta.
https://elifesciences.org/articles/55954#video1

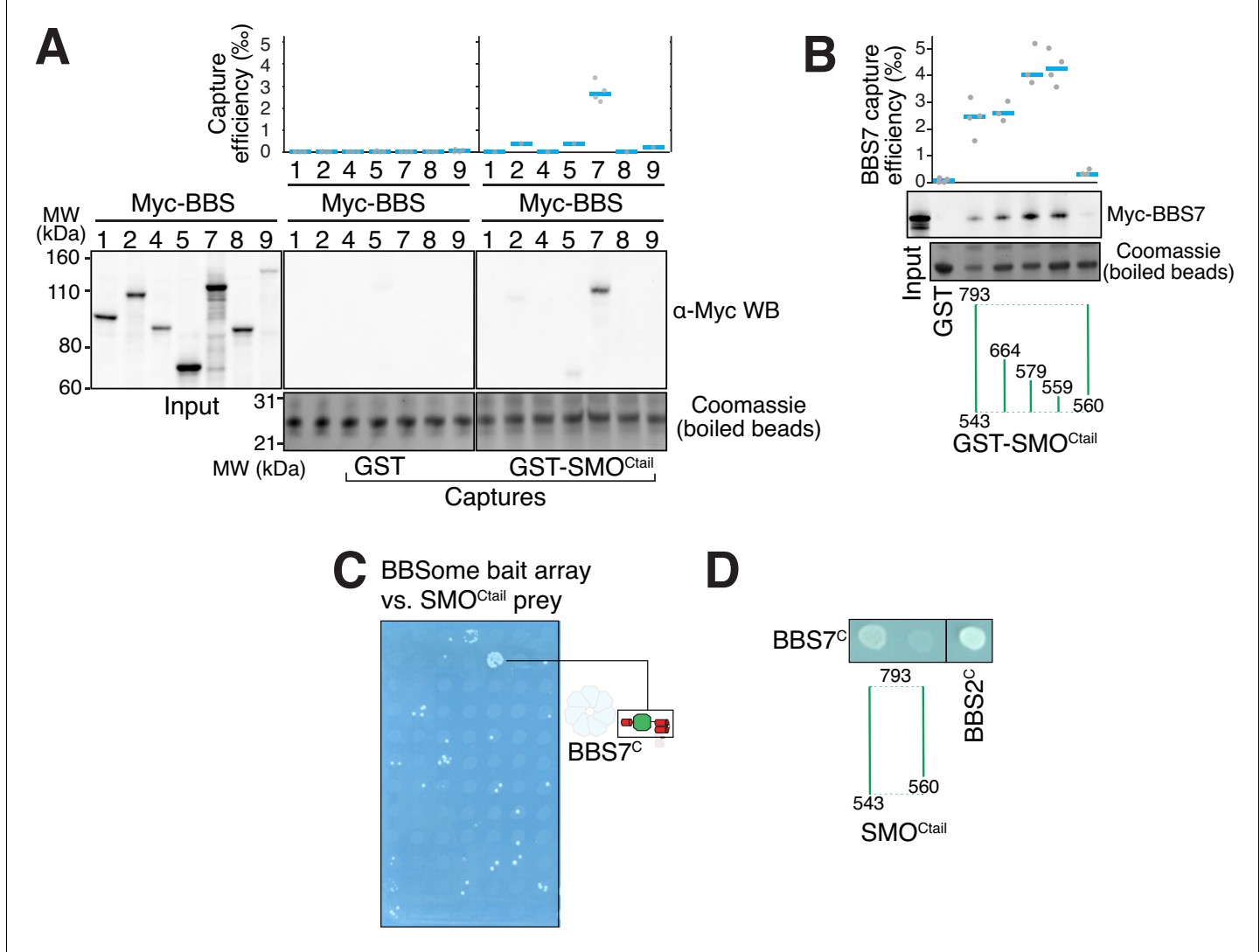

**Figure 3.** The BBSome recognizes SMO via membrane-embedded residues in SMO helix 8. (A–B) GST-capture assays were conducted with in vitro translated BBSome subunits tagged with a 6xMyc epitope. Bound material was released by specific cleavage between GST and the fused peptide, and released proteins were detected using a Western blot and anti-Myc antibody (α-Myc WB). The proportions of BBSome subunits recovered in the eluate are plotted; grey circles are individual data points and blue lines are mean values. Even loading of the glutathione beads is demonstrated by staining for the remaining GST-tagged proteins after cleavage elution. (A) Capture of individual BBSome subunits with GST-SMO$^{Ctail}$ (aa 543–793) identifies BBS7 as the SMO-binding subunit. (B) Capture assays with truncations of SMO$^{Ctail}$ find that SMO$^{H8}$ is necessary and sufficient for binding to BBS7. (C) Yeast two-hybrid (YTH) assays with SMO$^{Ctail}$ against an array of BBS protein fragments identify an interaction between a C-terminal fragment of BBS7 (BBS7$^C$, residues 326–672) and SMO$^{Ctail}$. The composition of the BBS YTH array is shown in *Supplementary file 2*. (D) YTH assays find that SMO$^{H8}$ is required for the interaction with BBS7$^C$. Growth controls on diploid-selective medium for panels (C–D) are shown in *Figure 3—figure supplement 1C*. The online version of this article includes the following figure supplement(s) for figure 3:

**Figure supplement 1.** Capture assays, controls for YTH and SMO$^{H8}$ conformation, and BBSome-binding motif in ciliary GPCRs.

complex with the BBSome is similar to that of other Arf-like GTPases in complex with coat adaptor complexes (*Figure 5—figure supplement 1B*; *Cherfils, 2014*). The BBSome–ARL6$^{GTP}$ complex displays a convex membrane-facing surface, defined by the N terminus of ARL6$^{GTP}$ and parts of BBS2$^{pf}$, BBS7$^{βprop}$ and BBS9$^{βprop}$, that espouses the contour of the ciliary membrane modeled as a 250 nm cylinder (*Figure 5C*). A convex membrane-binding surface in the Golgin$^{GRIP}$–ARL1$^{GTP}$ or MKLP1–ARF6$^{GTP}$ complexes similarly allows these complexes to associate with concave surfaces (*Makyio et al., 2012*; *Panic et al., 2003*). Importantly, a hydrophobic cluster traced through the

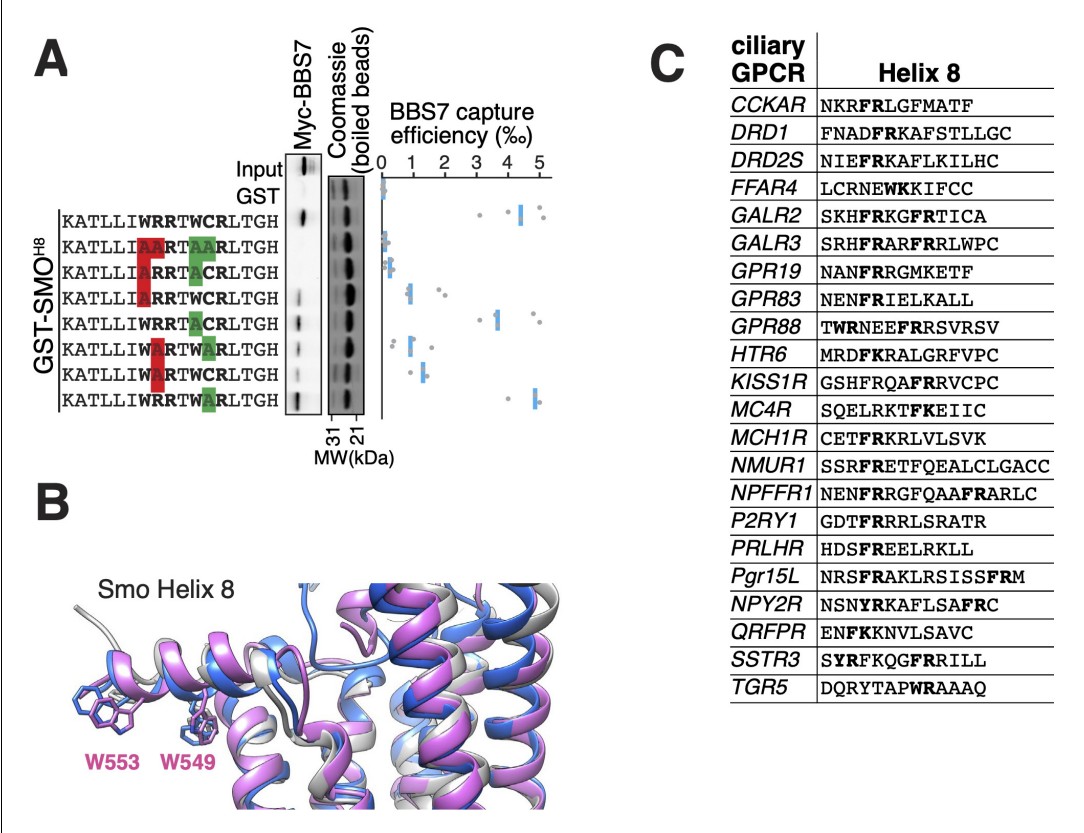

**Figure 4.** The BBSome BBS7 interacts with conserved SMO helix 8. (**A**) Capture assays of BBS7 with mutants of SMO[H8] (aa 543–559) identify Trp549 and Trp553 as the major BBS7-binding determinants of SMO[H8]. The sequences of the peptides fused to GST after the protease cleavage site are indicated. (**B**) Overlay of helix 8 from three structures of human SMO (PDB IDs: 5L7D, 6O3C, 6D32), showing that the orientation of the two tryptophan residues into the hydrophobic core of the membrane is conserved (see additional structures in *Figure 3—figure supplement 1D*). For consistency with the GST fusions used in capture assays, residue numbering corresponds to mouse SMO. (**C**) Sequence analysis finds a BBSome-binding motif ([W/F/Y]R) within helix 8 in 20 of the 26 GPCRs known to localize to cilia. Sequences are listed in *Supplementary file 3*.

surfaces of BBS7[cc] and BBS7[βprop] reveal a strong candidate for sheltering the critical Trp residues in SMO[H8] (*Figure 5D*).

## Molecular interactions of the BBSome with membranes and the IFT-B complex

Liposome-recruitment assays with pure BBSome and ARL6[GTP] have shown that the BBSome recognizes lipid headgroups, in particular the phosphoinositide PI(3,4)$P_2$ (*Jin et al., 2010*). Pleckstrin Homology (PH) domains are prototypical PIP-recognition modules and PIP-overlay assays suggested that BBS5[PH1] might directly recognize PIPs (*Nachury et al., 2007*), although it has been noted that PIP-overlay assays can report spurious interactions (*Yu et al., 2004*). We sought to determine whether the PH domains of BBS5 can bind to lipid headgroups in our model of the membrane- and cargo-bound BBSome. The canonical PIP-binding motif K$x_n$[K/R]xR is present in the β1-β2 loop of nearly all PH domains that bind PIPs (*Isakoff et al., 1998*; *Vonkova et al., 2015*). BBS5[PH1] contains a perfect match to the PIP-binding motif (K41xxxxxR47xR49) but no such motif is found in BBS5[PH2] (*Figure 6A*). Consistent with the absence of a PIP-binding motif in BBS5[PH2], lipid binding is blocked by the edge of a blade from BBS9[βprop] (*Figure 6B*). When the canonical PIP-binding site was mapped to the structure of BBS5[PH1], the lipid-binding site was occluded by a loop connecting BBS7[βprop] to BBS9[cc] (*Figure 6B*). Modeling 9 distinct PH domains co-crystallized with PIP headgroups onto BBS5[PH1] showed limited variance in the lipid orientation (*Figure 6C*). In summary, the PH domains of BBS5 are unable to recognize PIP through their canonical sites.

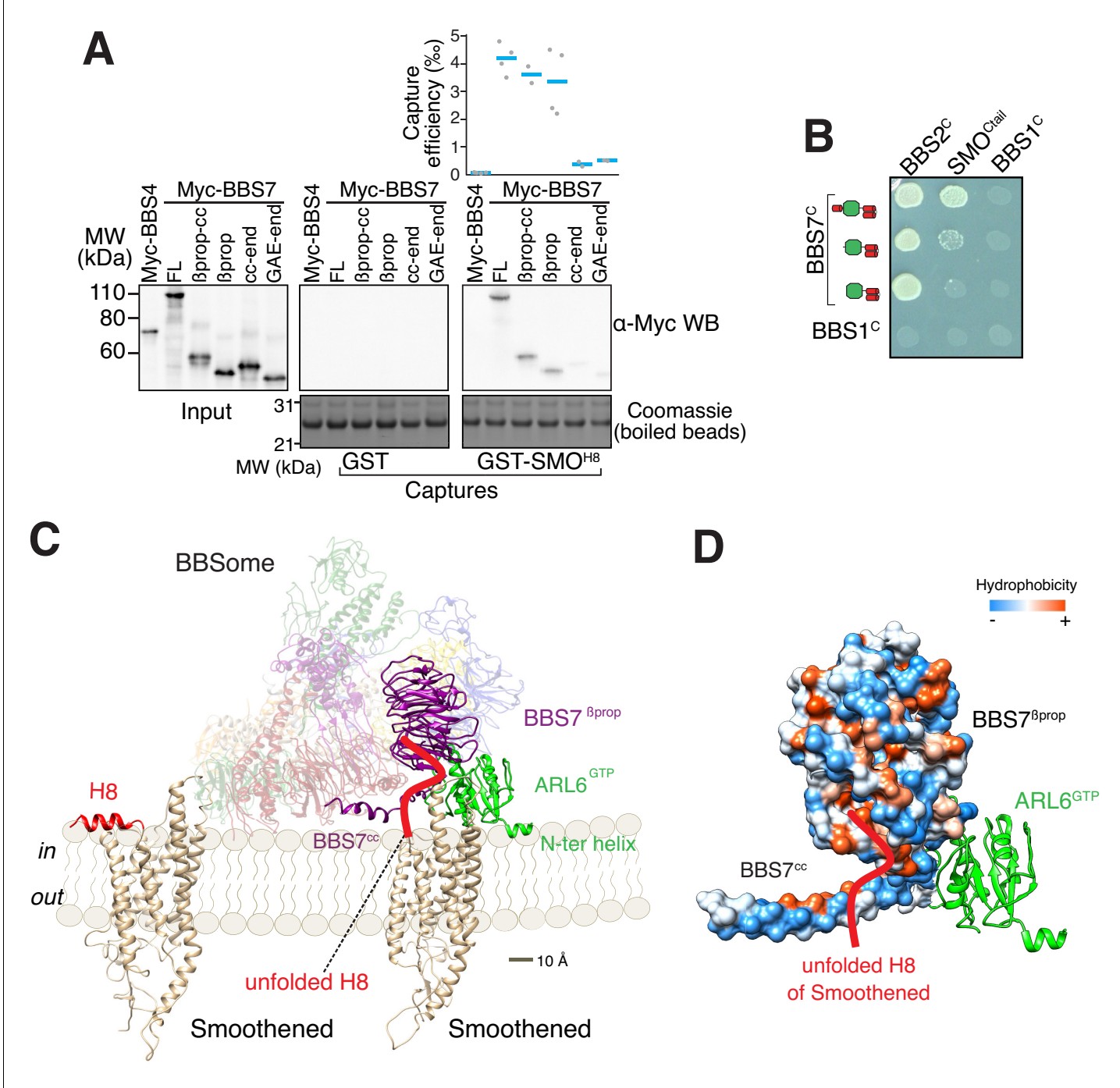

**Figure 5.** A model for binding of the BBSome to membranes and cargo. (**A**) Capture assays of BBS7 find that BBS7$^{\beta prop}$ engages SMO$^{H8}$. The boundaries of each truncation are βprop, aa 1–332; βprop-cc, aa 1–378; cc-end, aa 326–672; and GAE-end, aa 375–672. Results are presented as in *Figure 3*. (**B**) YTH assays show that the deletion of BBS7$^{cc}$ impairs the interaction of BBS7$^{C}$ with SMO$^{Ctail}$ (top row), but not with a C-terminal fragment of BBS2 (BBS2$^{C}$, residues 324–712) (middle rows). BBS1$^{C}$ serves as non-interacting control. Growth controls on diploid-selective medium are shown in *Figure 5—figure supplement 1A*. (**C**) Diagram illustrating the proposed interaction of SMO with the membrane-bound BBSome–ARL6$^{GTP}$ complex. For clarity, ARL6$^{GTP}$ and the BBS7 domains involved in SMO binding (BBS7$^{\beta prop}$ and BBS7$^{cc}$) are shown in solid colors with the remaining subunits shown with reduced opacity. Helix 8 (H8) of SMO is folded in the absence of partners, and is proposed to become a random coil in the SMO–BBSome complex. (**D**) Hydrophobicity surface of the BBS7$^{\beta prop}$ and BBS7$^{cc}$ domains, showing a plausible binding cleft for unfolded SMO$^{H8}$ (shown in red). ARL6$^{GTP}$ is shown in ribbon representation.

The online version of this article includes the following figure supplement(s) for figure 5:

**Figure supplement 1.** Controls for YTH and membrane orientation of ARF-like GTPases in complex with trafficking complexes.

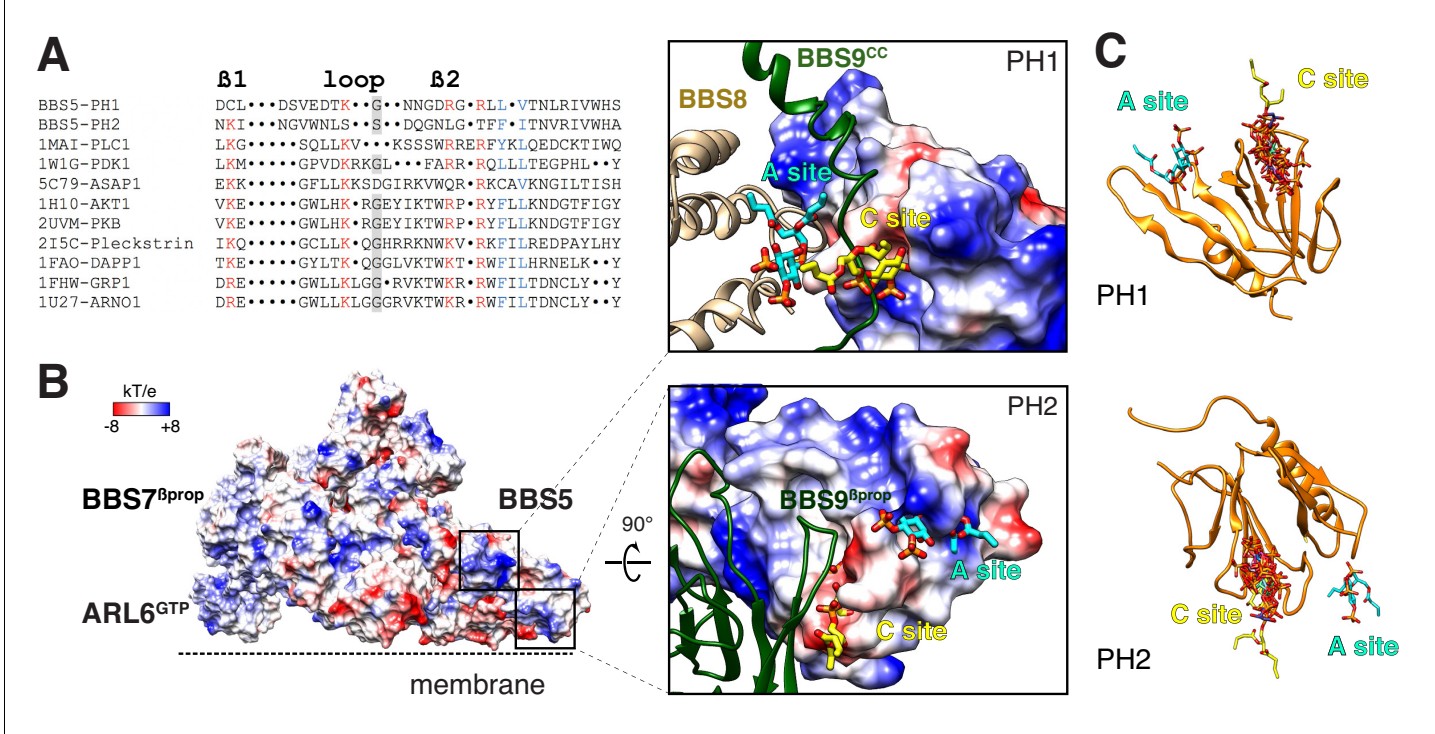

**Figure 6.** Mapping the putative interactions of lipids onto the BBSome structure. (A) Sequence alignment of the β1-loop-β2 region in structurally characterized PH domains. Conserved residues are colored: grey shading, glycine; red, positively charged residues; blue, hydrophobic residues. (B) Electrostatic surface of the membrane-bound BBSome–ARL6$^{GTP}$ complex, and close-up views of the BBS5 pleckstrin homology (PH) domains. For the PH1 domain, the canonical (C) and atypical (A) sites for lipid binding are occluded by BBS9 (dark green ribbon) and BBS8 (gold ribbon), respectively. For the PH2 domain, the C site is blocked by BBS9, but the A site is accessible. The lipids at the A and C sites, shown as yellow and cyan sticks, respectively, are diC4-PtdIns(4,5)P2 and are modeled based on the structural alignment of the BBS5 PH domains with the lipid-bound ASAP1 PH domain (PDB ID: 5C79) (C) Structural overlay of the lipids bound to the PH domains listed in the above sequence alignment (PDB IDs: 1MAI, 1W1G, 5C79, 1H10, 2UVM, 2I5C, 1FAO, 1FHW, 1U27), showing the consistency of the lipid position for both PH domains. The lipids are shown as stick models and are overlaid on the PH1 and PH2 domains of BBS5 (orange ribbon).

The online version of this article includes the following figure supplement(s) for figure 6:

**Figure supplement 1.** Proposed binding surfaces for membranes on the BBSome.

More recently, the crystal structure of the PH domain of ASAP1 has provided singular evidence for an atypical PIP-binding (A) site (*Jian et al., 2015*). In BBS5$^{PH1}$, lipid binding to the predicted A site extensively clashes with BBS8$^{TPR8-9}$ (*Figure 6B*). In BBS5$^{PH2}$, lipid binding to the putative A site would cause no steric clash (*Figure 6B*). However, considering that the distance between the membrane and the A site of BBS5$^{PH2}$ exceeds 1 nm and considering the limited evidence for the existence of A sites in PH domains, it is very unlikely that the A site of BBS5$^{PH2}$ participates in lipid binding of the BBSome.

Besides the PH domains of BBS5, we inspected the membrane-facing surface of the BBSome for positively charged surfaces (*Figure 6—figure supplement 1A*). Interestingly, the surface of the ARL6$^{GTP}$-bound BBSome exhibited considerably fewer negative charges facing the membrane than the surface of BBSome alone (compare *Figure 6—figure supplement 1A and B*). While some negatively charged surfaces facing the membrane remained, the surfaces closest to the membrane are generally positively charged in the BBSome–ARL6$^{GTP}$ complex structure.

We next considered binding of the BBSome to IFT. IFT38/CLUAP1 is the only IFT-B subunit to consistently interact with the BBSome in systematic affinity purification studies (*Boldt et al., 2016*) and a recent study found that the C terminus of IFT38 interacts with the BBSome in visual immunoprecipitation (VIP) assays (*Nozaki et al., 2019*). Using GST-capture assays with pure BBSome, we confirmed that IFT38 directly interacts with the BBSome and that the C-terminal domain of IFT38 is necessary and sufficient for this interaction (*Figure 7A*). The C-terminal tail of IFT38 (aa 329–413) is

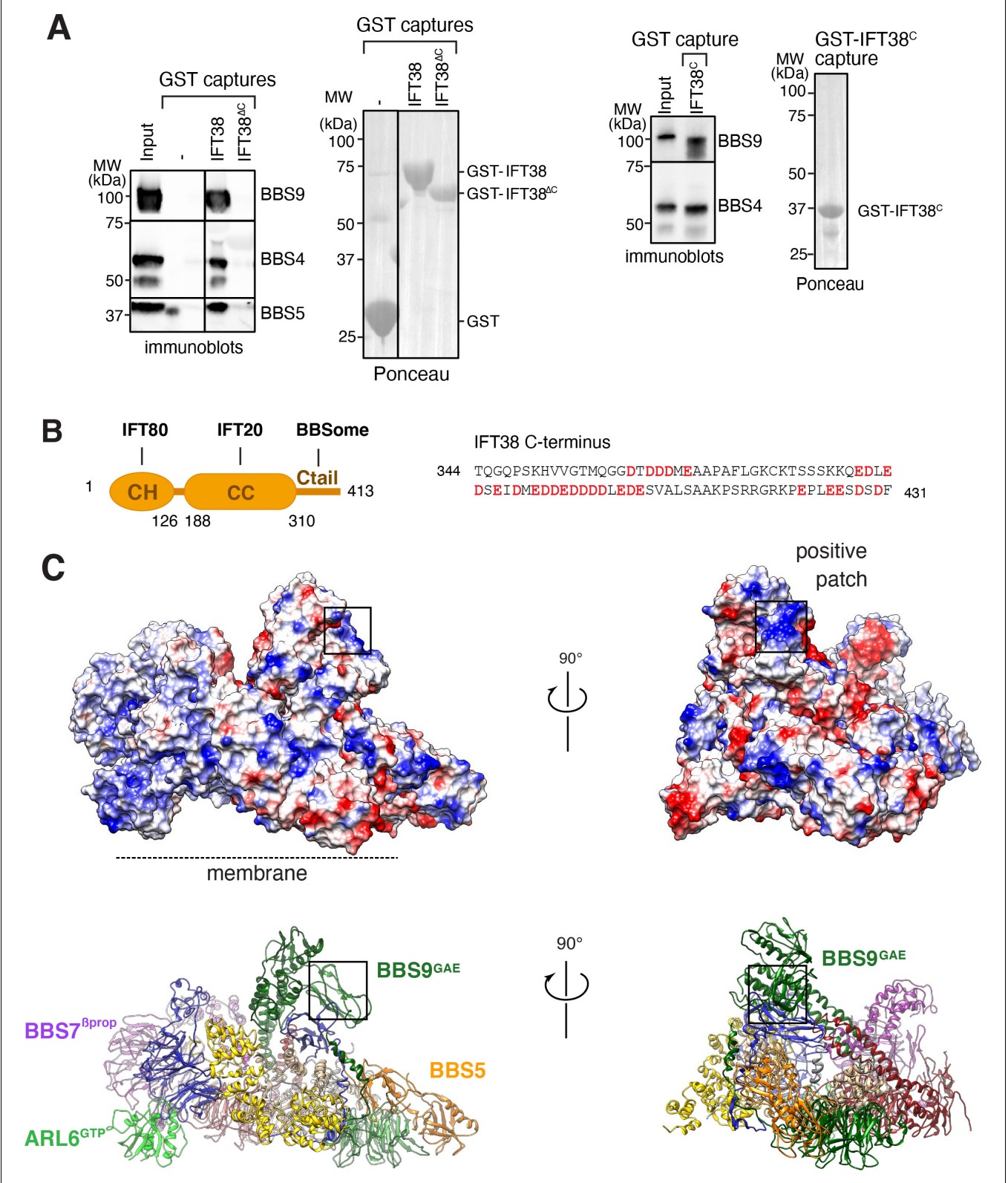

**Figure 7.** Proposed binding surfaces for IFT38 and membranes on the BBSome. (**A**) IFT38^Ctail is necessary and sufficient for BBSome binding. GST-capture assays were conducted with BBSome purified from bovine retina and GST fusions immobilized on glutathione sepharose. Bound material was eluted in SDS sample buffer. 2.5 input equivalents were loaded in the capture lanes. The BBSome was detected by immunoblotting and the GST fusions by Ponceau S staining. (**B**) Left panel: Diagram of the domain organization of IFT38. The calponin homology (CH) domain interacts with IFT80,

*Figure 7 continued on next page*

*Figure 7 continued*

the coiled-coil (CC) domain with IFT20, and the C-terminal tail (Ctail) with the BBSome. Right panel: Sequence of the region of IFT38 that interacts with the BBSome. Negatively charged residues are bold and red. (C) Top panels: Electrostatic surface of the ARL6[GTP]-bound BBSome showing a patch of positive charges on a region of BBS9 that interacts with BBS1 and BBS2 and is a candidate for binding IFT38[C]. Bottom panels: Corresponding orientations in ribbon diagram representations.

predicted to be unstructured and is extremely acidic with 30 Glu or Asp in its 85 amino acids, giving it a theoretical pI of 4.03 (*Figure 7B*). As VIP assays identified BBS9 as the major binding subunit of IFT38[C], with contributions from BBS2 and BBS1, we reasoned that a BBS9 domain in close proximity to BBS2 and BBS1 should be responsible for IFT38[C] binding. The C terminus of BBS9 (GAE, pf, hp, CtH) sits atop BBS1[GAE] and connects to BBS2[hp] and an extended positive patch is found in BBS9[GAE] (*Figure 7C*). This positive patch is therefore a strong candidate for the IFT38[C] interaction and its orientation away from the membrane makes it well-positioned to interact with IFT trains associated with the axoneme.

## Discussion

### Biochemical properties of the open conformation

With the mapping of the cargo-binding and the IFT38-binding surfaces on the BBSome, and the near-atomic structures of the closed and open conformation, we sought to determine whether these interactions are gated by the activating conformational change.

First, IFT38 interacts with the BBSome in systematic immuno-precipitation/mass spectrometry studies, in VIP assays, and in GST capture of pure BBSome. The candidate interacting region is diametrically opposite from the ARL6-binding region and does not undergo any measurable change upon conformational opening.

Second, BBS7[βprop] and BBS7[cc] appear equally accessible in the closed and open conformation (*Figures 1A* and *2A* and *Video 1*), suggesting that the conformational change is unlikely to directly increase the affinity of BBSome for its cargo SMO. This would contrasts with findings of increased affinity of COPI, AP1 and AP2 for their cargoes upon conformational changes induced by membrane recruitment (*Dodonova et al., 2017*; *Jackson et al., 2010*; *Ren et al., 2013*). It should nonetheless be noted that the densities for BBS7[βprop] and BBS7[cc] are much better defined in the ARL6[GTP]-bound structure (*Figure 1—figure supplement 2C*), thus suggesting that these regions become less mobile upon ARL6[GTP] binding. This change in mobility of BBS7[βprop] and BBS7[cc] are consistent with the direct binding of the backside of ARL6 to the BBS7[βprop]-BBS7[cc] connector loop. ARL6[GTP] binding to the BBSome may thus lock the cargo-binding determinants of the BBSome in an orientation that is optimal for cargo binding.

We conclude that the biochemical interaction that is modulated by ARL6[GTP] binding is likely distinct from cargo or IFT binding. More sensitive biochemical assays or the discovery of novel interactions may be necessary to decipher the interactions that are modulated by conformational opening of the BBSome.

Surprisingly, deletion of the BBSome-binding domain from IFT38 in cells did not grossly alter BBSome distribution in cilia or affect the ability of the BBSome to constitutively remove SMO from cilia, but it did interfere with GPR161 exit (*Nozaki et al., 2019*). These results suggest that IFT38 assists the BBSome with a subset of its duties rather than in cohesion between BBSome and IFT trains. It thus remains conceivable that the interaction of the BBSome with IFT-B is gated by ARL6[GTP]. In support of this hypothesis, we found that the recruitment of BBSome to large retrograde trains depends on ARL6 (*Ye et al., 2018*). More recently, (*Xue et al., 2020*) reported that ARL6[GTP] captures the BBSome together with the IFT-B complex from *Chlamydomonas* extracts while ARL6[GDP] only captures the BBSome. This finding suggests that, in *Chlamydomonas*, the interaction between the backside of ARL6 and the BBSome is sufficiently strong to produce stable binding. More importantly, these results indicate that a BBSome–IFT-B supercomplex may be specifically assembled in the presence of ARL6[GTP].

## Interaction of the BBSome with smoothened and other ciliary GPCRs

All our binding studies consistently identify BBS7 as the BBSome subunit responsible for the binding to SMO H8 (*Figure 3*). We note the caveat that we identified this interaction based on in vitro binding experiments using individual BBSome proteins and fragments. Future experiments will aim at validating this interaction by binding studies using fully assembled wild-type and mutant BBSomes, but such experiments will only become possible once the BBSome can be produced recombinantly. Our identification of BBS7 as the predominant cargo-binding subunit is in apparent conflict with a previous study that found that a core BBSome missing BBS7 still binds cargo (*Klink et al., 2017*). However, earlier co-IP studies found that the SMO C-tail interacts both with BBS7 and BBS5 (*Seo et al., 2011*). Notably, while our own GST/IVT-capture assays unambiguously showed that SMO C-tail interacts with BBS7, we also observe a weak band for BBS5 in these capture assays (*Figure 3A* and *Figure 3—figure supplement 1B*). It is thus possible that the cargo binding observed for the core BBSome missing BBS7 is based on interactions mediated by BBS5.

Our results show that the BBSome can only interact with SMO if its amphipathic H8 is extracted from the membrane. We investigated whether this binding requirement is generalizable to other GPCRs besides SMO. Searching for the BBSome-binding motif [W/F/Y][K/R] (*Klink et al., 2017*) within H8 of the 26 known ciliary GPCRs revealed that 23 of them contained a BBSome-binding motif in their helix 8 (*Figure 4C*). Considering that aromatic residues will point towards the core of the membrane in these H8, the broad distribution of BBSome-binding motifs suggests that the BBSome may bind to extracted H8 in nearly all ciliary GPCRs.

Because of its amphipathic nature, it will be a rare event for H8 to leave the membrane and be available for capture by the BBSome. It is thus noteworthy that the region of the BBSome involved in SMO C-tail binding is located directly adjacent to its ARL6-binding site (*Figure 5C,D*). The close proximity of the ARL6-binding site that tethers the BBSome to the membrane will ensure that the binding site for the SMO C-tail is also close to the membrane and will thus be in the best position to bind H8 as soon as it leaves the membrane.

SMO$^{H8}$ is predicted to be capped by a palmitoylated cysteine (Cys554), similar to helix 8 in class A GPCRs, which are frequently palmitoylated near their C termini (*Piscitelli et al., 2015*). More generally, a striking feature common to nearly all BBSome cargoes is the presence of palmitoyl and/or myristoyl anchors (*Liu and Lechtreck, 2018*). The BBSome must thus shelter a considerable hydrophobic surface when it extracts helix 8 of GPCRs out of the membrane. In the case of SMO, the BBSome has to stabilize the two tryptophan residues in SMO$^{H8}$ that normally reside in the hydrophobic core of the lipid bilayer (*Figure 3* and *Klink et al., 2017*) as well as the palmitoylated Cys554, which will find itself outside of the hydrophobic core of the membrane when SMO$^{H8}$ is bound to the BBSome. Therefore, the BBSome likely contains a cavity that shelters large hydrophobic residues and lipid anchors to hold helix 8 away from the membrane. Our structures reveal that binding of ARL6$^{GTP}$ results in the formation of a cleft in the BBSome that is close to BBS7$^{βprop}$ and BBS7$^{cc}$ (*Figure 5D*), the region involved in SMO$^{H8}$ binding, and is thus a potential location for the predicted cavity needed to shelter the SMO C-tail.

# Materials and methods

## Plasmid DNA

The plasmids for SP6-driven in vitro transcription of individual BBSome subunits are based on pCS2+Myc6-DEST vectors and were described in *Jin et al., 2010*. C-terminal truncations of BBS7 were generated by introducing stop codons using site-directed mutagenesis in pCS2+Myc6-BBS7. N-terminal truncations of BBS7 were assembled by PCR and Gateway recombination.

The plasmids for bacterial expression of SMO$^{ctail}$, SMO$^{H8}$ and IFT38 truncations are derivatives of pGEX6P1.

## Antibodies

Primary antibodies against the following proteins were used: actin (rabbit, Sigma-Aldrich, #A2066), cMyc (mouse, 9E10, Santa Cruz sc-40), acetylated tubulin (mouse, 6-11B-1, Sigma-Aldrich), ARL6 (rabbit, *Jin et al., 2010*), SMO (rabbit, gift from Kathryn Anderson, Memorial Sloan Kettering Cancer Center, New York, NY, *Ocbina et al., 2011*).

Secondary antibodies for immunoblotting were: HRP-conjugated goat anti-mouse IgG (115-035-003, Jackson Immunoresearch) and HRP-conjugated goat anti-rabbit IgG (111-035-003, Jackson Immunoresearch).

## Sequence analysis

Helix 8 sequences were collected from GPCRdb (https://gpcrdb.org, *Pándy-Szekeres et al., 2018*) and manually searched for BBSome-binding motifs. Ciliary GPCRs were collected from the literature (*Badgandi et al., 2017*; *Berbari et al., 2008*; *Hilgendorf et al., 2019*; *Koemeter-Cox et al., 2014*; *Loktev and Jackson, 2013*; *Marley et al., 2013*; *Marley and von Zastrow, 2010*; *Mukhopadhyay et al., 2013*; *Omori et al., 2015*; *Siljee et al., 2018*).

## Recombinant protein expression

N-terminally GST-tagged ARL6ΔN16[Q73L] was expressed in bacteria as described (*Chou et al., 2019*). GST-tagged SMO$^{Ctail}$ and IFT38 protein fusions were expressed in Rosetta2(DE3)-pLysS cells grown in 2xYT medium (Millipore Sigma, Y2627) at 37°C until the optical density (OD) at 600 nm reached 0.6. Protein expression was then induced with 1 mM isopropyl β-D-1-thiogalactopyranoside (IPTG) at 18°C for 4 hr (SMO$^{Ctail}$) or with 0.2 mM IPTG at 18°C for 16 hr (IFT38). Cells were resuspended in 4XT (200 mM Tris, pH 8.0, 800 mM NaCl, 1 mM DTT) with protease inhibitors (1 mM AEBSF, 0.8 µM Aprotinin, 15 µM E-64, 10 µg/mL Bestatin, 10 µg/mL Pepstatin A and 10 µg/mL Leupeptin) and lysed by sonication. The clarified lysates were loaded onto Glutathione Sepharose 4B resin (GE Healthcare) and proteins eluted with 50 mM reduced glutathione in buffer XT (50 mM Tris, pH 8.0, 200 mM NaCl, 1 mM DTT). Proteins were subsequently dialyzed against XT buffer with one change of buffer and flash frozen in liquid nitrogen after addition of 5% (w/v) glycerol.

## Purification of native BBSome

The BBSome was purified from bovine retina by ARL6$^{GTP}$-affinity chromatography as described (*Chou et al., 2019*) and the sample was processed for cryo-EM the next day.

## GST-capture assays

GST pull-down assays were conducted by saturating 10 µL of Glutathione Sepharose 4B beads (GE #17075605) with GST fusions. Binding to purified BBSome was assessed by mixing beads with a 10 nM solution of pure BBSome made in 100 µL IB buffer (20 mM HEPES, pH 7.0, 5 mM MgCl$_2$, 1 mM EDTA, 2% glycerol, 300 mM KOAc, 1 mM DTT, 0.2% Triton X-100) and incubating for 1 hr at 4°C. After 4 washes with 200 µL IB buffer, elution was performed by boiling the beads in SDS sample buffer.

BBSome subunits and fragments thereof were translated in vitro from pCS2-Myc plasmids using the TNT SP6 Quick Coupled Transcription/Translation system (Promega L2080). 16 µL TNT SP6 Quick Master Mix, 2 µL Methionine (0.2 mM) and 2 µL DNA (0.2 µg/µL) were mixed and incubated at 30°C for 90 min. 20 µL reactions were diluted into 180 µL NSC250 buffer (25 mM Tris, pH 8.0, 250 mM KCl, 5 mM MgCl$_2$, 0.5% CHAPS, 1 mM DTT), mixed with 10 µL glutathione beads saturated with GST fusions and rotated for 1 hr at 4°C. After 4 washes with 200 µL NSC250 buffer, elution was performed by addition of 7.5 µg PreScission protease in 30 µL NSC250 buffer and incubation at 4°C overnight. The eluates were resolved by SDS-PAGE and analyzed by immunoblotting with anti-Myc antibody.

## Yeast two-hybrid assays

The coding DNA sequences (CDSs) for various fragments of BBSome subunits were either obtained in Gateway Entry vectors or amplified via PCR and transferred to pDONR221 by BP clonase recombination. The CDSs were shuttled to Y2H Gateway destination vectors bait pBTMcc24 (C-terminal bait), pBTM116D-9 (N-terminal bait), pCBDU (C-terminal prey), and pACT4 (N-terminal prey) by LR clonase recombination. Bait and prey vectors were introduced into either bait (L40ccU MATa) or prey (L40ccα MATα) yeast strains by lithium acetate transformation. Yeast were mated in a 96-well matrix format, using at least two independently transformed colonies to test each interaction. MATa and MATα yeast were mated on YPDA medium for 36–48 hr at 30°C prior to diploid selection on medium lacking tryptophan and leucine. Diploids were incubated for 2 days at 30°C prior to transfer

onto medium lacking tryptophan, leucine and histidine to select for positive growth of interacting constructs.

## Cryo-EM sample preparation and data collection

For BBSome alone, 3.5 µL of the peak fraction from a BBSome purification (0.4–0.6 mg/mL) was applied to glow-discharged R1.2/1.3 holey carbon copper grids (Quantifoil) covered with a thin homemade carbon film. The grids were blotted for 1 s at 4°C and 100% humidity, and plunged into liquid ethane using a Mark IV Vitrobot (Thermo Fisher Scientific). A cryo-EM dataset was collected on a 300-kV Titan Krios electron microscope (Thermo Fisher Scientific) equipped with a K2 Summit detector (Gatan) at a nominal magnification of 22,500x in super-resolution counting mode. After binning over $2 \times 2$ pixels, the calibrated pixel size was 1.3 Å on the specimen level. Exposures of 10 s were dose-fractionated into 40 frames with a dose rate of 8 e⁻/pixel/s, resulting in a total dose of 80 e⁻/Å². Data were collected with SerialEM (*Mastronarde, 2005*) and the used defocus range was from −1.5 µm to −3.0 µm.

For the BBSome–ARL6 complex, full-length ARL6 was incubated with GTP at a molar ratio of 1:20 for 1 hr on ice, added to purified BBSome at a molar ratio of 5:1 and incubated for another hour on ice. 3.5 µL of the sample was applied to glow-discharged R1.2/1.3 holey carbon grids (Quantifoil Au or C-flat Cu). The grids were blotted for 3.5 s at 4°C and 100% humidity, and plunged into liquid ethane using a Mark IV Vitrobot. One cryo-EM dataset was collected on a 300-kV Titan Krios electron microscope equipped with a K2 Summit detector at a nominal magnification of 28,000x in super-resolution counting mode. After binning over $2 \times 2$ pixels, the calibrated pixel size was 1.0 Å on the specimen level. Exposures of 10 s were dose-fractionated into 40 frames with a dose rate of 7.52 e⁻/pixel/s, resulting in a total dose of 75.2 e⁻/Å². A second dataset was collected on a 300-kV Titan Krios equipped with a K3 detector at a nominal magnification of 64,000x in super-resolution counting mode. After binning over $2 \times 2$ pixels, the calibrated pixel size was 1.08 Å on the specimen level. Exposures of 2 s were dose-fractionated into 50 frames with a dose rate of 29.99 e⁻/pixel/s, resulting in a total dose of 51.44 e⁻/Å². Both datasets were collected with SerialEM and the defocus ranged from −1.5 µm to −2.5 µm.

## Cryo-EM data processing

The movie frames collected with the K2 detector were corrected with a gain reference. All movies were dose-weighted and motion-corrected with MotionCor2 (*Zheng et al., 2017*). The contrast transfer function (CTF) parameters were estimated with CTFFIND4 (*Rohou and Grigorieff, 2015*, p. 4). For micrographs collected with the K2 detector, particles were picked with Gautomatch (https://www.mrc-lmb.cam.ac.uk/kzhang/Gautomatch/); for those collected with the K3 detector, particles were picked with RELION 3.0 (*Zivanov et al., 2018*). Three projections from our previous cryo-EM map of the BBSome (EMD-7839) were used as templates for picking.

For BBSome alone, 2,218,320 particles were picked from 4733 micrographs and subjected to 2D classification in RELION. Particles in classes that generated averages showing clear structural features were selected (770,345 particles) for ab initio 3D reconstruction of two models in cryoSPARC (*Punjani et al., 2017*). The map with clearer structural features and higher resolution was selected for heterogenous refinement in cryoSPARC, after which 560,777 particles were selected for further homogenous refinement. The output map was further refined in RELION by 3D refinement, CTF refinement and Bayesian polishing, resulting in a map at 3.6 Å resolution. The base and corkscrew modules of the BBSome, including BBS1, BBS4, BBS5, BBS8, BBS9 and BBS18, were well resolved, but density for the top lobe, containing BBS2 and BBS7, was weak. A focused refinement, masking out the BBS1 βprop and ins, BBS2 βprop, GAE and cc, and the BBS7 βprop, GAE, cc, pf and hp domains, yielded a map for the remainder of the BBSome at 3.44 Å resolution.

For the BBSome–ARL6 complex, data collected with the K2 camera yielded 228,487 particles from 1031 micrographs of a Quantifoil Au grid and 192,243 particles from 911 images of a C-flat Cu grid. The particles from the two datasets were separately subjected to 2D classification in RELION, and particles from classes that generated averages showing clear structural features were combined, including 134,169 and 72,182 particles, respectively. Data collected with the K3 camera yielded 1,033,939 particles from 2680 micrographs of the Quantifoil Au grid and 688,499 particles from 1960 micrographs of the C-flat grid. After 2D classification, 185,332 and 154,503 particles,

respectively, were selected. All selected particles were combined (546,186 particles in total) and subjected to 3D classification into 6 classes, using as reference the previously determined BBSome map (EMD-7839) filtered to 45 Å resolution. One of the resulting maps showed clear fine structural features (209,646 particles) and was subjected to 3D refinement, yielding a density map at 4.1 Å resolution. Refinement focused on the top lobe of the BBSome, including BBS2, BBS7, BBS1 βprop, and ARL6 yielded a map at 4.2 Å resolution. Refinement focused on the lower lobe of the BBSome including the remaining subunits yielded a map at 3.8 Å resolution. To improve the density for the GAE and pf domains of BBS2 and BBS7, a mask was generated for these domains and used for focused 3D classification into 4 classes without alignment. The resulting map showing the best structural features was selected for further refinement, which resulted in a map at 4.0 Å resolution, with improved density for the GAE and pf domains of BBS2 and BBS7.

The resolution was determined by Fourier shell correlation (FSC) of two independently refined half-maps using the 0.143 cut-off criterion (*Rosenthal and Henderson, 2003*). Local resolution was estimated from the two half-maps using the ResMap algorithm implemented in RELION. UCSF Chimera (*Pettersen et al., 2004*) was used to visualize density maps. Statistics are listed in *Supplementary file 1*.

## Model building and refinement

Our previously published backbone model of the BBSome (*Chou et al., 2019*) was first placed into the density map using Chimera. All manual model building was performed with Coot (*Emsley and Cowtan, 2004*). BBS2$^{GAE}$ and BBS7$^{GAE}$ models were generated using SWISS-MODEL (*Waterhouse et al., 2018*), using the structure of BBS9$^{GAE}$ as template. The generated models were then docked into the density map using Chimera, and trimmed in Coot. Due to the weak density of these areas in both maps, we only built secondary-structure fragments but not the connecting loops. A model for bovine ARL6 starting was generated with SWISS-MODEL, using the crystal structure of *Chlamydomonas reinhardtii* ARL6 (PDB ID: 4O0VN) as template. The model was then docked into the density map of the BBSome–ARL6 complex. The atomic models were refined using phenix.real_space_refine (*Adams et al., 2010*). Cryo-EM data collection, refinement and modelling statistics are summarized in *Supplementary file 1* Table S1.

## Acknowledgements

We thank Mark Ebrahim and Johanna Sotiris for help with grid screening and data collection at the Evelyn Gruss Lipper Cryo-Electron Microscopy Resource Center at The Rockefeller University. This work was funded by NIGMS (R01-GM089933, MVN), NEI (R01- EY031462, MVN and TW) and Research to Prevent Blindness (Stein Innovator Award A131667, MVN). This work was made possible, in part, by NEI (EY002162 - Core Grant for Vision Research, MVN), RPB (Unrestricted Grant, MVN) and the Austrian Science Fund (FWF P30162, US). Molecular graphics and analyses were performed with the UCSF Chimera package. Chimera is developed by the RBVI at UCSF (supported by NIGMS P41-GM103311).

## Additional information

### Funding

| Funder | Grant reference number | Author |
| --- | --- | --- |
| National Institute of General Medical Sciences | R01-GM089933 | Maxence V Nachury |
| Research to Prevent Blindness | Stein Innovator Award A131667 | Maxence V Nachury |
| National Eye Institute | R01- EY031462 | Thomas Walz Maxence V Nachury |

The funders had no role in study design, data collection and interpretation, or the decision to submit the work for publication.

## Author contributions

Shuang Yang, Conceptualization, Data curation, Formal analysis, Investigation, Methodology, Writing - original draft, Writing - review and editing; Kriti Bahl, Hui-Ting Chou, Investigation; Jonathan Woodsmith, Formal analysis, Investigation; Ulrich Stelzl, Supervision; Thomas Walz, Supervision, Writing - original draft, Writing - review and editing; Maxence V Nachury, Conceptualization, Formal analysis, Supervision, Funding acquisition, Investigation, Writing - original draft, Writing - review and editing

## Author ORCIDs

Shuang Yang https://orcid.org/0000-0001-8566-5092
Jonathan Woodsmith http://orcid.org/0000-0002-0790-3726
Ulrich Stelzl http://orcid.org/0000-0003-2500-3585
Thomas Walz http://orcid.org/0000-0003-2606-2835
Maxence V Nachury https://orcid.org/0000-0003-4918-1562

## Decision letter and Author response

Decision letter https://doi.org/10.7554/eLife.55954.sa1
Author response https://doi.org/10.7554/eLife.55954.sa2

# Additional files

## Supplementary files

- Supplementary file 1. Cryo-EM data collection, refinement and modelling statistics.
- Supplementary file 2. Fragments used in the BBS YTH array displayed in *Figure 3C*.
- Supplementary file 3. Sequences of helix 8 from known ciliary GPCRs.
- Transparent reporting form

## Data availability

Structural data have been deposited into the Worldwide Protein Data Bank (wwPDB) and the Electron Microscopy Data Bank (EMDB). The EM density map for the BBSome has been deposited under accession code EMD-21251 and the EM density map for the BBSome-ARL6 complex has been deposited under accession code EMD-21259. The corresponding atomic models have been deposited under accession codes 6VNW and 6VOA.

The following datasets were generated:

| Author(s) | Year | Dataset title | Dataset URL | Database and Identifier |
|---|---|---|---|---|
| Yang S, Bahl K, Chou H-T, Nachury MV, Walz T | 2020 | Cryo-EM structure of apo-BBSome | http://www.ebi.ac.uk/pdbe/entry/emdb/EMD-21251 | Electron Microscopy Data Bank, EMD-21251 |
| Yang S, Bahl K, Chou H-T, Nachury MV, Walz T | 2020 | Cryo-EM structure of apo-BBSome | http://www.rcsb.org/structure/6VNW | RCSB Protein Data Bank, 6VNW |
| Yang S, Bahl K, Chou H-T, Nachury MV, Walz T | 2020 | Cryo-EM structure of the BBSome-ARL6 complex | http://www.ebi.ac.uk/pdbe/entry/emdb/EMD-21259 | Electron Microscopy Data Bank, EMD-21259 |
| Yang S, Bahl K, Chou H-T, Nachury MV, Walz T | 2020 | Cryo-EM structure of the BBSome-ARL6 complex | http://www.rcsb.org/structure/6VOA | RCSB Protein Data Bank, 6VOA |

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
