## [Decision Letter]

**Acceptance summary:**

The BBSome complex ferries membrane receptors out of the cilia. This paper reports high resolution structures of the BBSome with and without the membrane targeting G-protein Arl6 complementing two other recent structures. Interaction data with the BBSome cargo smoothened (SMO) suggests an exciting new model in which a C-terminal amphipathic helix of SMO must be released from the membrane to allow for BBSome binding.

**Decision letter after peer review:**

Thank you for submitting your article "Near-atomic structures of the BBSome reveal a novel mechanism for transition zone crossing" for consideration by *eLife*. Your article has been reviewed by three peer reviewers, and the evaluation has been overseen by Andrew Carter as Reviewing Editor and Anna Akhmanova as the Senior Editor. The following individual involved in review of your submission has agreed to reveal their identity: Esben Lorentzen (Reviewer #2).

In summary, the reviewers were all strongly supportive of the publication of your structural work. They had some concerns about your binding studies showing Smo^H8^ binds BBS7, although felt that this could be addressed. Their main concern was with the last section of the paper in which you propose that the TZ curvature blocks Smo H8 exit, requiring it to bind to the BBSome. They thought the model was interesting but currently the data behind it are too preliminary.

The suggestion is that you revise the manuscript to remove the last section and address the concerns regarding the Smo H8 binding. I include a summary of the major comments below.

Summary:

The manuscript by Bahl et al., reports the high resolution single particle Cryo-EM structures of the BBSome complex with and without the small GTPase ARL6 bound. This study significantly extends a previous reported BBSome structure at lower resolution by the same labs and complements the other recent structures of the BBsome. In addition, the authors carry our binding studies of individual proteins of the BBSome with the amphipathic helix 8 of the SMO cargo and present a model where SMO helix 8 must be released from the membrane to allow for BBSome binding and ciliary exit. Furthermore, the authors map the binding between the IFT subunit IFT38 and the BBSome and analyse the biophysical properties of the ARL6 amphipathic helix necessary for ciliary exit of BBSomes.

Essential revisions:

1) Binding assays with BBSome proteins show that it is BBS7 only that binds Smo H8, that Smo W549, R550, and W553 are important for binding, and that BBS7^cc^ and BBS7-prop are the parts of the BBSome that bind Smo H8. The results are solid, but the conclusions they allow are somewhat limited. The authors tested the ability of all of the BBSome proteins to bind Smo c-tail, and only BBS7 did. They then did a series of additional assays to arrive at the conclusions about the residues in Smo H8 and in BBS7 that are responsible for the in vitro binding they detected. These results are sufficient to allow them to propose that these interactions also might function in vivo. To publish these results, though, the manuscript must emphasize their limitations.

a) Ideally the reviewers would like to see experiments performed using intact BBSomes to show that BBS7 binds the Smo c-tail in the context of the full BBSome. Due to the current difficulties in doing lab work they would be content with changes to the manuscript indicating that future Smo H8 binding experiments are required using intact WT BBSomes and BBSomes containing BBS7b with mutations designed to disrupt Smo H8 binding to establish whether the results translate to the full BBSome.

b) Second you need to directly indicate that Klink et al., reported that the "core BBSome" lacking BBS7 binds Smo c-tail and also propose reasonable, testable explanations for the discrepancies. The reviewers feel you could do more to link the structural results to the experiments showing that BBS7 binds Smo. For example: are there reasons that the site on the BBSome that binds Smo must be near the site on the BBSome that is close to Arl6 and thus close to the membrane?

2) Currently the study feels like two independent studies (structural and functional) that were merged together without building on the resulting discoveries collectively. For example, could you discuss whether Arl6^GTP^ binding to BBSome plays a role in allowing the BBS7 propeller to recruit SMO H8 out of the membrane?

---

## [Author Response]

Essential revisions:1) Binding assays with BBSome proteins show that it is BBS7 only that binds Smo H8, that Smo W549, R550, and W553 are important for binding, and that BBS7^cc^ and BBS7^prop^ are the parts of the BBSome that bind Smo H8. The results are solid, but the conclusions they allow are somewhat limited. The authors tested the ability of all of the BBSome proteins to bind Smo c-tail, and only BBS7 did. They then did a series of additional assays to arrive at the conclusions about the residues in Smo H8 and in BBS7 that are responsible for the in vitro binding they detected. These results are sufficient to allow them to propose that these interactions also might function in vivo. To publish these results, though, the manuscript must emphasize their limitations.a) Ideally the reviewers would like to see experiments performed using intact BBSomes to show that BBS7 binds the Smo c-tail in the context of the full BBSome. Due to the current difficulties in doing lab work they would be content with changes to the manuscript indicating that future Smo H8 binding experiments are required using intact WT BBSomes and BBSomes containing BBS7b with mutations designed to disrupt Smo H8 binding to establish whether the results translate to the full BBSome.

We agree with the reviewers that binding assays would ideally be performed with fully assembled wild-type and mutant BBSomes. However, such experiments will require the expression of recombinant BBSomes, which – despite substantial efforts by us and other groups – has not yet been possible. We therefore added the following statement to the Discussion in our revised manuscript: “We note the caveat that we identified this interaction based on in vitro binding experiments using individual BBSome proteins and fragments. Future experiments will aim at validating this interaction by binding studies using fully assembled wild-type and mutant BBSomes, but such experiments will only become possible once the BBSome can be produced recombinantly.”

b) Second you need to directly indicate that Klink et al., reported that the "core BBSome" lacking BBS7 binds Smo c-tail and also propose reasonable, testable explanations for the discrepancies.

We now explicitly state that Klink et al. reported that their core BBSome lacking BBS7 binds SMO C-tail. We also discuss that a previous study found by co-IP that SMO C-tail interacts with both BBS7 and BBS5 (Seo et al., 2011). In addition, our capture assay with in vitro-translated BBSome subunits showed unambiguous capture of BBS7 but also a faint band for BBS5. It is therefore possible that Klink et al. detected the interaction of SMO C-tail with BBS5 in their core BBSome with SMO C-tail.

The reviewers feel you could do more to link the structural results to the experiments showing that BBS7 binds Smo. For example: are there reasons that the site on the BBSome that binds Smo must be near the site on the BBSome that is close to Arl6 and thus close to the membrane?

We thank the reviewers for this comment. While we have no experimental evidence, we added the following speculation to the revised manuscript: residue Trp549, part of the BBSome-binding motif, is very close to the end of the 7^th^ transmembrane helix of Smoothened. For most of the time Trp549 will be embedded in the membrane and will only occasionally be exposed to the solvent. The proximity of the ARL6-binding site to the SMO-binding site may thus increase the probability that the SMO-binding site is close to the membrane and will be able to capture the SMO C-tail as it temporarily leaves the membrane.

2) Currently the study feels like two independent studies (structural and functional) that were merged together without building on the resulting discoveries collectively. For example, could you discuss whether Arl6^GTP^ binding to BBSome plays a role in allowing the BBS7 propeller to recruit SMO H8 out of the membrane?

The established function of ARL6^GTP^ is to recruit the BBSome to the ciliary membrane, where Smoothened is located. As discussed above, ARL6^GTP^ may also keep the SMO-binding site close to the membrane to increase the likelihood that it captures the C-tail when it leaves the membrane. Furthermore, the BBSome should contain a cavity that shelters the large hydrophobic residues and lipid anchors to stabilize unfolded helix 8 away from the membrane. Our structures show that ARL6^GTP^ binding results in the formation of a cleft in the BBSome that is close to BBS7^βprop^ and BBS7^cc^ and is thus a potential location for the predicted cavity that shelters the SMO C-tail. We added this idea to the Discussion.